# The Role of Tumor Microenvironment in the Pathogenesis of Sézary Syndrome

**DOI:** 10.3390/ijms23020936

**Published:** 2022-01-15

**Authors:** Denis Miyashiro, Bruno de Castro e Souza, Marina Passos Torrealba, Kelly Cristina Gomes Manfrere, Maria Notomi Sato, José Antonio Sanches

**Affiliations:** 1Division of Clinical Dermatology, University of São Paulo Medical School, Sao Paulo 05403-900, Brazil; brunocastro1990@hotmail.com (B.d.C.e.S.); jasanches@usp.br (J.A.S.); 2Instituto do Câncer do Estado de São Paulo, University of São Paulo Medical School, Sao Paulo 01246-000, Brazil; 3Laboratory of Medical Investigation, LIM-56, Department of Dermatology, Tropical Medicine Institute of São Paulo, University of São Paulo Medical School, Sao Paulo 05403-000, Brazil; marinatorrealba@usp.br (M.P.T.); kellycgmanfrere@usp.br (K.C.G.M.); marisato@usp.br (M.N.S.)

**Keywords:** Sézary syndrome, cutaneous T-cell lymphoma, tumor microenvironment, cytokines, chemokines

## Abstract

Sézary syndrome is an aggressive leukemic variant of cutaneous T-cell lymphomas, characterized by erythroderma, lymphadenopathy, and peripheral blood involvement by CD4+ malignant T-cells. The pathogenesis of Sézary syndrome is not fully understood. However, the course of the disease is strongly influenced by the tumor microenvironment, which is altered by a combination of cytokines, chemokines, and growth factors. The crosstalk between malignant and reactive cells affects the immunologic response against tumor cells causing immune dysregulation. This review focuses on the interaction of malignant Sézary cells and the tumor microenvironment.

## 1. Introduction

Sézary syndrome (SS) was first described by Albert Sézary and Yves Bouvrain in 1938 [1]. It is a rare and aggressive leukemic variant of cutaneous T-cell lymphoma (CTCL). Males are more affected than females (2:1), and it occurs almost exclusively in adults. The classic triad of SS includes erythroderma, lymphadenopathy, and circulating malignant cells [2]. Besides erythroderma, a diffuse non-scarring alopecia, palmoplantar hyperkeratosis, nail dystrophies, and leonine facies may be observed [3]. Intense pruritus is the most frequent symptom, and it significantly decreases the quality of life [2,4]. Systemic symptoms (fever, night sweats, and weight loss) are present in 1.6% of the patients [5]. Circulating Sézary cells are detected by peripheral blood smear (large lymphocytes with cerebriform nuclei) and immunophenotyping of lymphocytes by flow cytometry (CD4:CD8 ≥ 10, CD4+CD7− ≥ 40%, CD4+CD26− ≥ 30%). A search for the T-cell receptor (TCR) gene rearrangement shows a monoclonal population of T-cells on the blood, and the exact clone is detected on skin infiltrate (Figure 1) [2,6,7].

The pathophysiology of SS is not entirely understood. The most plausible hypothesis is the activation of T-cells by antigen-presenting cells, leading to the gradual accumulation of mutations that culminates with neoplastic cell development. However, the triggering antigen is unknown, and it could vary between patients [8,9].

Mycosis fungoides (MF) and SS were considered the same disease for many years. However, neoplastic cells in these two entities have distinct origins. MF cells strongly express C-C chemokine receptor (CCR)-4 and cutaneous lymphocyte-associated antigen (CLA), which confer tropism to the skin, and are negative for CCR7 and L-selectin, receptors that confer tropism to the lymph nodes. This immunophenotype is characteristic of skin-resident memory T-cells. On the other hand, Sézary cells express CCR7 and L-selectin, CD27 (a characteristic marker of central memory T-cells), CCR4, and other skin-tropic receptors (CCR6, CCR10, CLA). These findings suggest that MF and SS originate from different subtypes of T lymphocytes [10].

The prognosis of SS is poor. The five-year overall survival rates range between 40 and 50% [2,11]. First-line treatment includes extracorporeal photopheresis (ECP), interferon-α combined with ECP or phototherapy, retinoids, chlorambucil associated with prednisone, and low dose methotrexate. Second-line treatment includes chemotherapy with gemcitabine, pegylated liposomal doxorubicin, CHOP (cyclophosphamide, doxorubicin, vincristine, and prednisone), and CHOP-like regimens, alemtuzumab (anti-CD52 monoclonal antibody), and allogeneic stem cell transplantation [12].

Response to chemotherapy is excellent; however, the disease rapidly recurs. Biologic response modifiers (ECP, interferon-α, and retinoids) show prolonged responses, suggesting an essential role of immunomodulation in controlling neoplastic proliferation. Increasing evidence shows the influence of the tumor microenvironment characterized by reactive cells, chemokines, and cytokines in the clinical behavior of SS, and these interactions gradually drive an antitumor response towards a tolerogenic milieu [13,14]. Understanding the tumor microenvironment is essential to the development of new therapies to improve survival and, maybe, reach a curative treatment for this aggressive and morbid disease. This review focuses on the discussion of the current scenario of the immunologic milieu of SS.

## 2. Tumor Microenvironment

The interaction between tumor cells and the microenvironment influences the progression of CTCL. In early-stage MF, neoplastic cells are scarce and reactive T-helper (Th) 1 and CD8+ T-lymphocytes contribute to the antitumor defense [15,16,17]. With the disease’s progression, the tumor microenvironment shifts from a Th1 to a Th2 response, and contributes to tumor cells growth and immune escape [15,18].

As an aggressive CTCL, SS is a Th2-type disease, and it exhibits an exhaustion status of antitumor defense [15,19,20], with increased levels of interleukin (IL)-4, IL-5, and IL-13, and reduced levels of Th1 cytokines such as IL-2 and interferon (IFN)-γ, that reduce cell-mediated immunity [18,21]. This microenvironment favors angiogenesis, tissue remodeling, as well as survival and proliferation of malignant cells. The Th2 cytokines produced by Sézary cells suppress the Th1 response and impair cellular immunity. Reactive T-cells are present but are dysfunctional due to the Th1/Th2 imbalance [22,23].

It is not known precisely how malignant cells start this change in the tumor microenvironment. It is hypothesized that somatic mutations, somatic copy number variations, and epigenetic deregulation in Sézary cells could drive the activation of pro-oncogenic and the inhibition of tumor suppressor pathways [24,25]. However, exogenous factors, mainly *Staphylococcus aureus* colonization, play an important role in the emergence of the Th2 response [26]. The endogenous and exogenous factors will ultimately affect the JAK/STAT pathway, with a decrease in STAT4 (Th1) and an increase in STAT3, STAT5, and STAT6 (Th2) activation in neoplastic cells [27]. Under normal conditions, STAT proteins are transiently activated. However, in neoplastic cells, constitutive activation of STAT3, STAT5, and STAT6 occur. Aberrant activation of these transcription factors stimulates the secretion of Th2 cytokines. The Th1 transcription factor STAT4 is inhibited in SS, probably due to the action of micro-RNA (miR)-155 induced by STAT5; [15,28]. and GATA-3, a Th2 transcription factor, is activated through STAT6 signaling [29]. The released Th2 cytokines contribute to a positive feedback loop between malignant and reactive cells, influencing the growth and survival of the former [30].

### 2.1. CD8+ T-Cells

The CD8+ T-cells are cytotoxic lymphocytes that play an important role in antitumor response by exocytosis of intracytoplasmic granules with perforin, granzymes, and T-cell-restricted intracellular antigen-1 (TIA-1), and by a Fas-mediated pathway in which membrane-bound Fas ligand (FasL) expressed on CD8+ T-cells interacts with Fas on neoplastic cells [31]. The intensity of CD8+ T-cell infiltrate within the skin of CTCL patients is associated with a better prognosis [31].

In SS, circulating CD8+ T-cells express CD38, PD-1, Tim-3, and CD39. The CD38 is an activation marker frequently observed in chronic viral infections, and PD-1, Tim-3, and CD39 are exhaustion markers (Figure 2). T-cell exhaustion is a state of dysfunction observed in chronic infections and cancer due to the persistence of antigens and inflammation. After persistent antigen stimulation, CD8+ T-cells undergo functional loss [13]. Their cytotoxicity is modified, and their cytokine production ability, proliferative capacity, and effective memory cell generation are also affected. Impaired production of IFN-γ, tumor necrosis factor (TNF), IL-2, and a high expression of coinhibitory receptors are observed, which compromises their ability to fight against infections and tumor cells [19,32,33].

The IL-7 and IL-15 are growth factors essential to lymphocytic functions. The IL-7 induces IFN-γ production and proliferation; the IL-15 contributes to the proliferation and survival of natural killer (NK) and CD8+ T-cells. Upon binding to their receptors on the cell surface, they signal via JAK1 and JAK3, which activate STAT5 [19,34,35]. In CD8+ T-cells, STAT5 induces the antiapoptotic molecule Bcl-2 expression, which is important to maintain CD8+ effector function [36,37,38]. In SS, CD8+ T-cells exhibit impaired STAT5 and Bcl-2 expression compared to healthy donors, even after IL-7 stimulus. These cells also exhibit increased CD95/Fas expression, which may trigger apoptosis, contributing to the decreased cytotoxic activity against neoplastic cells [19].

### 2.2. Regulatory T-Cells

The Tregs comprise five to ten percent of peripheral T-cells [39]. These cells inhibit other T-cell functions by secretion of inhibitory cytokines such as IL-10 and transforming growth factor (TGF)-β; by induction of apoptosis mediated through secretion of granzymes A/B and perforin; by expression of tumor-necrosis-factor-related-apoptosis-inducing ligand—death receptor 5 (TRAIL-DR5); by induction of Fas/FasL pathway, galectin-9 pathway, and galectin-1 secretion; by metabolic disruption, due to the metabolism of ATP to AMP and the production of adenosine (an immunoregulatory purine) and transfer of cyclic AMP to effector cells by gap junctions that lead to apoptosis by IL-2 deprivation; and by modulation of dendritic cell (DC) maturation or function through the interaction of CTLA-4 on Tregs with its ligand CD80/86 on antigen-presenting cells [40,41].

Sézary cells commonly express FoxP3 and CD25, markers observed in Tregs, and the global methylation pattern of Sézary cells are similar to the one observed in Tregs. Thus, it has been postulated that SS may represent a malignancy of Tregs in some patients [42,43,44,45]. Importantly, FoxP3 and CD25 in malignant cells do not seem to exclusively confer a suppressive phenotype. Some patients may present high levels of FoxP3 and CD25 in Sézary cells but not in other patients, in which, malignant T-cells failed to suppress T-cell proliferation [46]. There is evidence of low molecular splice forms of FoxP3 that are functionally different from wild type FoxP3 and not involved in the execution of the suppressive function [47]. Some factors, such as cytokines or bacterial toxins in the tumor microenvironment have been proposed to drive the heterogeneous FoxP3 expression in malignant cells [48]. The Treg properties expressed by Sézary cells are stimulated by staphylococcal enterotoxins (SEs), which trigger the expression of FoxP3 in a STAT5-dependent manner, and by direct contact with immature DCs via MHC class 2 presentation of antigens of apoptotic cells. These Sézary cells with Treg properties secrete IL-10 and TGF-β, which suppress the secretion of IL-2 and IFN-γ and maintain DC immaturity, contributing to further neoplastic cell proliferation and up-regulation of the Treg phenotype [23,39,49].

Programmed death-1 (PD-1) is an immune checkpoint inhibitor. It is increased in benign and malignant CD4+ T-cells in SS. When activated, the PD-1 axis inhibits reactive T-cells, promotes the induction of Th2 and Treg cells, and prevents apoptosis of Tregs. Thus, nonmalignant Tregs are increased in SS, contributing to the immune escape of tumor cells [21,50,51].

### 2.3. Regulatory B-Cells

A subset of B-cells can suppress immune responses, similar to Tregs, called regulatory B-cells (Bregs). These cells contribute to immune tolerance by secretion of IL-10 but also other inhibitory molecules, including PD-L1, granzyme B, TGF-β, and IL-35, leading to the induction of tumor immunosuppressive cells [52]. Few studies addressing the presence of Bregs in CTCL are available. Interestingly, decreased Bregs are observed in CTCL progression, and it is hypothesized that Bregs suppress the activity of tumor cells in the blood [14,53]. Other B-cells are also decreased in CTCL, such as CD19+ CD24hiCD27+ B-cells, CD19+ CD38hi B-cells, together with IL-10-producing B-cells in CTCL progression [53]. Despite the discussion that Breg may play a role in the CTCL progression, IL-10 produced by Bregs enriched in CD19+ CD24hiCD27+ B-cells could impair the function of immune cells, including Th1/CD8+/NK cells, or TGF-*β* secreted by Bregs could convert CD4+  T-cells into Tregs that would promote tumor progression [54]. More studies are needed to understand whether Bregs can affect other immune cells or help to convert malignant cells to Tregs in Sézary syndrome.

### 2.4. NK Cells

The NK cells are cytotoxic lymphoid cells that constitute the innate immune system [55]. The NK cells can be divided into two main populations: CD56^dim^CD16^bright^, which is predominant in the peripheral blood and has a cytolytic activity; and CD56^bright^CD16^dim^, which is predominant on lymph nodes and secondary lymphoid tissue and has an immunoregulatory role [56].

The NKG2D is the main activating receptor of the NK cells, and it binds to major histocompatibility complex (MHC) class I homologs (MICA and MICB) and UL-16 binding proteins (ULPB)-1 to 5, that function as signals of cellular stress [57,58]. Upon binding to NKG2D ligand (NKG2DL) expressed in malignant cells, the NK cells attack by degranulation of perforin and granzyme and production of IFN-γ and TNF-α [14]. In a process called trogocytosis, the cell-to-cell contact allows the migration of the NKG2DL of the tumor cell to the NK cell. Thus, these altered NK cells may be recognized by tumor-naïve NK cells and are killed (Figure 3) [59]. Other mechanisms that favor immune escape by malignant cells, besides the trogocytosis, are the down-regulation of NKG2D observed in NK cells from SS patients [58], and the reduced number of total NK cells and the cytolytic CD56^dim^CD16^bright^ NK cells, while the immunoregulatory CD56^bright^CD16^dim^ NK cells are preserved [60]. The downregulation of NKG2D may be due to TGF-β and metalloproteinases [61]. The subset of CD57+NKG2C+ NK cells has been described as exhibiting memory-like features, with potent effector functions, and could be elicited by human cytomegalovirus (HCMV) infection [62]. Interestingly, besides the altered cytolytic CD56^dim^ NK cells in SS patients, an increased percentage of CD56+CD57+NKG2C+ NK cells was found, together with high seropositivity to CMV [58]. The expansion of this mature CD57+NKG2C+ NK subset detected in SS patients could be due to its memory for CMV infection. The fact that trained immunity may display potent functions could be beneficial for cancer patients [63].

### 2.5. Dendritic Cells

Dendritic cells (DCs) are professional antigen-presenting cells. They prompt immune responses by activating naïve T-cells at a mature state and promote tolerance by deleting self-reactive thymocytes, mediating the anergy of mature T-cells, and generating Tregs at an immature state [14]. The Th2 cytokines secreted in the SS tumor microenvironment suppress the maturation of DCs. The influence of dysregulated or immature antigen-presenting cells can explain aspects of tumor progression [18].

The IL-10 downregulates DC functions, contributing to the formation of immature DCs, that promote tolerance rather than immune defense because these cells present apoptotic cell antigens without the appropriate co-stimulation. Furthermore, direct contact with immature DCs stimulates the Treg phenotype in Sézary cells (Figure 4) [23,64,65,66]. An increased number of immature DCs in SS lesions is important for immunological tolerance against malignant T-cells [67]. Conversely, IFN-γ stimulates the maturation of DCs, which inhibit Sézary cells proliferation [65,68]. Mature DCs may attempt to mount an immune response against the cancer cells via the production of the Th1 cytokines IL-12, IL-2, and IFN-α, as mature DCs are elevated in the skin draining lymph nodes of some patients [69,70,71]. Another dendritic cell subset, termed plasmacytoid dendritic cells (pDCs) are highly effective in sensing intracellular viral or self DNA and RNA mainly via Toll-like receptors (TLRs) and rapidly producing large amounts of type I and III interferons (IFNs) [72]. SS patients demonstrate the gradual loss of plasmacytoid dendritic cells in the peripheral blood [71]. However, using synthetic oligodeoxynucleotides with CpG motifs (CpG ODN), an agonist of TLR9, is able to induce IFN-α production, by the CD123 pDC of patients with Sézary syndrome [70].

The observation that apoptotic neoplastic cells can induce antigen-presenting cell maturation explains why total skin electron beam and extracorporeal photopheresis, which induce massive apoptosis of malignant cells, are effective treatment regimens [23].

### 2.6. Myeloid-Derived Suppressor Cells

Myeloid-derived suppressor cells (MDSCs) are immature myeloid cells that mediate immunosuppression by JAK3/STAT3-dependent secretion of reactive oxygen species (ROS) and arginase-1 (ARG-1) into the tumor microenvironment [51,73]. These cells are generated under pathological conditions, through myelopoiesis in the bone marrow or spleen before migration to the periphery. Granulocyte-macrophage colony-stimulating factor (GM-CSF), granulocyte colony-stimulating factor (G-CSF), macrophage colony-stimulating factor (M-CSF), stem cell factor (SCF), vascular endothelial growth factor (VEGF), IL-6, S100A9, S100A8, and prostaglandin E2 (PGE2) mediate MDSCs formation. These molecules induce signaling pathways such as STAT3, STAT5, interferon regulatory factor (IRF)-8, and CCAAT/enhancer-binding protein (C/EBP)-β. The C-C ligand (CCL)-2 and CCL-5 chemokines mediate recruitment of MDSCs to tumor microenvironment by binding to a receptor present on MDSCs. Other chemokines that induce mobilization of MDSCs in the tumor microenvironment include CCL7, CCL15, CCL26, C-X-C ligand (CXCL)-8, and CXCL12 [74].

The ROS upregulation in MDSCs is mediated by nicotinamide adenine dinucleotide phosphate (NADPH) oxidase activity. ROS production has been associated with T-cell unresponsiveness and tolerance [75].

The L-arginine is a substrate for the inducible nitric oxide synthase (iNOS) and ARG-1, both expressed in MDSCs and involved in lymphocyte suppression. Deprivation of L-arginine leads to T-cell dysfunction [76]. The iNOS generates nitric oxide, which is implicated in the attenuation of MHC class II expression in macrophages and in inducing T-cell apoptosis [77]. Thus, MDSCs are correlated with disease progression and resistance to therapy in hematologic malignancies. Indoleamine 2,3-dioxygenase (IDO)-dependent tryptophan catabolism is a mechanism of immunosuppression mediated by MDSCs [74,78]. MDSCs downregulate the expression of L-selectin, a key homing receptor on T-cells [79], promote the generation of Tregs through the secretion of TGF-β, IL-10, and IDO, increase M2 macrophages and suppress NK function by inhibiting IFN-γ production, NKG2D expression, and cytotoxic activity [74].

In SS, the number of MDSCs is not increased [51]. ROS production by MDSC is increased in CTCL suggesting that MDSC activity, rather than absolute numbers in peripheral blood, may correlate with disease progression [51]. MDSCs are a heterogeneous population of immature *myeloid* cells that include *monocytic* (mMDSC) and *granulocytic* (gMDSC) subsets, whereas, there are few studies on Sézary syndrome, to better understand their role in the immunosuppression.

### 2.7. M2 Macrophages

Macrophages present in the tumor microenvironment are called tumor-associated macrophages (TAM), and they are characterized by the expression of CD163, a highly specific monocyte/macrophage marker for polarized M2 macrophages [64]. There are two subpopulations of macrophages. The M1 macrophages are present in Th1 responses. They are induced by IFN-γ, present antigens, and produce inflammatory cytokines such as IL-1β, TNF-α, IL-6, and IL-23, and are related to inflammation and tumor inhibition. On the other hand, M2 macrophages are part of the Th2 response, are induced by IL-4, produce IL-10 and TGF-β, and are related to an immunosuppressive microenvironment, which favors tumor cell growth, angiogenesis, matrix remodeling, and inhibition of adaptive immunity. The release of IL-32 by NK cells, T-cells, keratinocytes, and fibroblasts may increase the M2 population in CTCL [80]. CD163/CD68 ratio was the highest at the MF tumor stage and Sézary syndrome, indicating M2 polarization with disease progression [81].

The IL-4 and IL-13 Th2 cytokines induce the production of CCL18 by M2 macrophages. The CCL18 binds to CCR8, a receptor expressed by Th2, Treg cells, and eosinophils (Figure 5) [14,82,83]. TAMs producing CCL18 are observed in CTCL patients, and serum levels of CCL18 are associated with a poorer prognosis [64,84]. Furthermore, increased CCL22 serum levels and sCD163 serum levels in CTCL reflect the increased activity of TAMs and tumor progression to a more advanced stage [81]. A strong correlation between macrophage depletion and decreased expression of a vascular marker, CD31, and lymphatic marker, podoplanin, suggest a role for macrophages in angiogenesis. Xenografted human CTCL cells (Hut78) showed that M2-like macrophages have a role in the progression of tumor formation in the skin [85].

The phagocytic activity of macrophages is also impaired in SS. The Th2 microenvironment with increased IL-4, IL-7, and IL-13, contributes to the expression of CD47 by Sézary cells. The CD47 binds to its receptor, the signal regulatory protein α (SIRPα), on macrophages, and inhibits macrophage-mediated phagocytosis of neoplastic cells (do-not-eat-me signal) (Figure 6) [23,86]. The blocking of CD47 in hematological malignancies showed good responses in preliminary clinical trials, and it highlights the role of phagocytosis in controlling malignant cellular growth [87].

### 2.8. Neutrophils

Neutrophils are polymorphonucleate granulocytes. They are part of the innate immune system and act by the phagocytosis of pathogens. The recognition of the pathogens occurs by the interaction of toll-like receptors (TLRs) and pathogen-associated molecular patterns (PAMPs) [88].

Neutrophils have major effector mechanisms as phagocytosis, degranulation, and Neutrophil extracellular traps (NETs) formation. NETs are composed of decondensed nuclear or mitochondrial DNA enriched by proteases and various inflammatory mediators. Cancer cells recruit neutrophils releasing NETs to the tumor microenvironment [89]. There is no evidence regarding NET production in CTCL, whereas neutrophils in peripheral blood show an activated profile. Neutrophils showed increased CD11b and CD66b and decreased CD62L, consistent with neutrophil activation [90]. Peripheral blood neutrophils in CTCL patients showed an enhanced respiratory burst and have an activated surface marker phenotype, even in the early stages of CTCL.

The presence of CXCL8 that mediates neutrophil recruitment has been detected in CTCL skin lesions as well as by clonal CTCL cells [91,92]. The elevations in plasma IL-8 show a mechanism for systemic neutrophil priming and activation in CTCL.

The IL-17 is an important cytokine involved in the pathogenesis of many skin diseases, e.g., psoriasis and hidradenitis suppurativa [93,94]. The IL-17 upregulates the secretion of C-X-C receptor (CXCR)-2 ligands. Neutrophils express CXCR2 and are attracted by the Th17 microenvironment [95,96]. In CTCL, a relatively low expression of IL-17 is observed, which may explain the lack of neutrophil infiltration in skin lesions [97]. Besides the low number of neutrophils in SS, they are functionally impaired, with reduced phagocytic activity and intracellular killing. These defects favor the development of infections in these patients, and the impaired response against pathogens is an important cause of complications and death [23,88]. The expression of IL-17A and IL-17F in a JAK3/STAT3/STAT5-dependent mechanism has been observed in a subset of patients, and it has been associated with disease progression. However, these cells do not express other characteristic Th17 phenotypes, suggesting that the capacity to produce IL-17 derives from a dysregulated signaling rather than a true Th17 response [15,98].

### 2.9. Mast Cells

Mast cells are bone marrow-derived hematopoietic cells. They are preferentially located in the skin, airways, and gastrointestinal tract, tissues that are in direct contact with the environment [99,100]. Mast cells are activated by immunoglobulin E (IgE) upon its binding to the high-affinity IgE receptor present on the cell surface. After activation, mast cells secrete histamine, proteases, cytokines, and chemokines [101]. Mast cells produce different matrix metalloproteinases (e.g., MMP-9) and proteases (tryptase and chymase) and could be an important source of proangiogenic factors [102]. Mast cells have rapid sensing of microorganisms such as bacteria, parasites, fungi, and viruses, which can be recognized by TLRs, resulting in the signaling pathway for the release of multiple cytokines as well as the release of preformed granules [102].

The tumor microenvironment in solid and hematopoietic malignancies is influenced by mast cells. Its increase in neoplastic tissues is correlated with tumor stage, prognosis, and invasiveness in different malignancies [17].

In CTCL, an increased number of mast cells particularly at the periphery of tumors is also correlated with tumor microvessel density and disease progression [103]. In a mouse model with deficient mast cell mice, tumor growth significantly decreased [101]. Mast cells in CTCL tissue exhibit a degranulated phenotype, and supernatant from the activated mast cells is able to promote proliferation of the malignant CTCL cells in vitro, which shows the protumorigenic role of mast cells [101].

Mast cells and histamine may play a role in CTCL, particularly in the advanced stages of the disease [104]. Mast cells activation leads to proteinases and histamine secretion which in turn stimulates sensory nerve endings and activates keratinocytes [105]. Considering that mast cells can be regulated by neurotransmitters and neuropeptides, it is important to understand the pathophysiology of cutaneous lymphoma-associated pruritus in Sézary syndrome.

### 2.10. Eosinophils

Eosinophils are innate immune granulocytes derived from the bone marrow and are associated with helminthic infections, allergic diseases, and many inflammatory diseases (e.g., eosinophilic esophagitis, eosinophilic pneumonitis, and eosinophilic cellulitis) [106,107,108,109]. Eosinophils are recruited into sites of inflammation and release major basic protein, eosinophil cationic protein, eosinophil peroxide, eosinophil-derived neurotoxins, IL-4, IL-5, IL-13, and GM-CSF [14].

The CCL26 and CCL11 are produced by dermal fibroblasts, keratinocytes, and endothelial cells, and are increased in the skin of SS patients. The CCR3 is the receptor for CCL26 and CCL11, and it is expressed on eosinophils. Thus, the upregulation of CCL26 and CCL11 induce the migration of eosinophils to the skin [64,110]. The IL-5 and IL-13 Th2 cytokines present in the tumor microenvironment of SS patients also favor eosinophilia, and IL-4 increases IgE [23]. Activated eosinophils secrete more IL-4, IL-5, IL-13, and angiogenic factors. Thus, eosinophils contribute to tumor growth by increasing immune tolerance with a Th2 response and facilitating neovascularization [15,111]. Eosinophils produce vascular endothelial growth factors (VEGFs) and other pro-angiogenic cytokines known to play a role in tumor progression in cancer.

Eosinophils are in association with other myeloid cell types that stimulate tumor-promoting inflammation as Treg and M2 macrophages [112]. Moreover, STAT3 activation in T-cells with neoplastic morphology was significantly associated with the presence of eosinophils in CTCL. Malignant T-cells also expressed eosinophilic activation and trafficking factors, such as High-mobility group BOX-1 protein (HMGB1) and IL-5, suggesting that these cells orchestrate the accumulation and activation of eosinophils in CTCL [113]. Tissue eosinophil activation in CTCL might contribute to the inflammatory flare-ups associated with aggressive T-cell lymphomas [114]. Blood eosinophilia at baseline should be considered a prognostic factor of poor outcome in patients with CTCL [115]. Increased IL-5 production by peripheral mononuclear cells from patients with Sézary syndrome together with eosinophilia was identified, suggesting that IFN-alpha and perhaps IL-12 may produce a therapeutic response in patients with CTCL and eosinophilia through the direct suppression of IL-5 production by malignant Sézary cells [116].

### 2.11. Keratinocytes

Keratinocytes are the most abundant cells in the epidermis. They express toll-like receptors (TLRs) that are crucial to a Th1-type immune response with the production of interferon. Keratinocytes also express MHC class II and act as non-professional antigen-presenting cells [14].

Keratinocytes are an important source of chemokines that contribute to the skin-homing of neoplastic and inflammatory cells in CTCL. The CCL17 is expressed by keratinocytes, Langerhans cells, and endothelial cells in the skin of SS patients. It binds to CCR4, present in Sézary cells. The CCL27 is constitutively produced by keratinocytes, and the erythrodermic skin of SS patients presents an increased CCL27 production compared to healthy normal skin. The receptor CCR10 is expressed on Sézary cells. Thus, the CCL17-CCR4 and CCL27-CCR10 interactions are essential to skin-homing of Sézary cells (Figure 7) [110,117].

The CCL26 and CCL11 are also produced by keratinocytes, and they contribute to the migration of eosinophils by interaction with the CCR3, which favors the Th2 response (Figure 8) [64].

Periostin is an extracellular matrix protein secreted by dermal fibroblasts upon stimulation by IL-4 and IL-13. Periostin mediates thymic stromal protein (TSLP) production by keratinocytes, and TSLP subsequently activates immature DCs, which modulate Th2 immune responses via CCL17 production. Immature DCs produce IL-4, IL-5, and IL-13. Serum and plasma TSLP levels are elevated in SS [118]. TSLP also induces STAT5 activation that promotes CTCL cells proliferation and IL-4 and IL-13 production [30,64,119,120]. STAT5 also downregulates STAT4 and the transcription factor STAB1 (special AT-rich sequence binding protein-1) through the induction of miR-155. The STAB1 inhibits the expression of IL-5 and IL-9 in neoplastic cells, and STAT5 activation allows the expression of these cytokines favoring the Th2 response (Figure 9) [30,121]. Periostin also induces the expression of IL-25 in keratinocytes. IL-25 promotes a Th2 immune response by enhancing the expression of IL-13, and IL-13 promotes the proliferation of malignant cells [122,123].

In SS, IL-22 is increased, and it promotes CCL20 expression in keratinocytes and induces epidermal hyperplasia. The CCL20 ligand, CCR6, is expressed on immature DCs, Th17, Th22, and regulatory T-cells. The migration of immature DCs and Tregs due to the increased expression of CCL20 by keratinocytes contributes to the immunological tolerance microenvironment, especially seen in advanced cases [30,97].

The most important symptom in SS patients is pruritus. It is observed in virtually all patients with SS, and its intensity is directly correlated with a reduction in the quality of life [3,124]. Nerve growth factor (NGF) is produced by keratinocytes, stimulates nerve fibers growth, and is associated with the severity of pruritus. NGF serum is elevated in SS, and the enhanced expression may be associated with increased dermal nerve fibers density and severe pruritus [125].

Angiogenin is a stimulator of angiogenesis, and it also acts as an inhibitor of polymorphonuclear cell degranulation. It is produced by keratinocytes and endothelial cells, besides being elevated in SS skin, and may be related to an increased susceptibility to infections and poorer prognosis [126,127].

### 2.12. Fibroblasts

Fibroblasts are spindle-shaped cells that are responsible for the production of the structural and signaling molecules present in the extracellular matrix, e.g., collagens, proteoglycans, elastin, fibronectin, microfibrillar proteins, and laminins [128]. Cancer-associated fibroblasts (CAFs) are crucial components of the tumor microenvironment, inducing cell growth and immune escape through various mechanisms [129]. Dermal fibroblasts in advanced-stage CTCL contribute to a Th2-dominant microenvironment by increasing Th2 and attenuating Th1 immune responses [17].

In SS, fibroblasts contribute to the Th2 microenvironment by producing CCL26, which attract CCR3+ eosinophils (Figure 8); and by secreting periostin upon IL-4 and IL-13 stimulation, which will mediate TSLP production by keratinocytes and will activate immature DCs (Figure 9) [64].

The herpesvirus entry mediator (HVEM) is a member of the tumor necrosis factor receptor superfamily. It is expressed on dermal fibroblasts of early-stage CTCL patients. The HVEM increases the production of CXCL9, CXCL10, and CXCL11, which recruit CXCR3+ Th1 cells to the skin. In advanced-stage CTCL, including SS patients, HVEM is decreased, and it attenuates the Th1 response [130]. The CXCR3 may also be expressed by malignant T-cells, and a decreased production of CXCL9, CXCL10, and CXCL11 may contribute to the loss of epidermotropism observed in SS [15]. On the other hand, fibroblasts produce CXCL12, or stromal cell-derived factor-1 (SDF-1). It is a chemoattractant for CXCR4+ tumoral cells and it is increased in the skin of SS patients. The CXCL12/SDF-1 is inhibited by CD26 peptidase activity. Since Sézary cells lack CD26, the CXCR4-CXCL12/SDF-1 axis may have an important role in the skin recruitment and accumulation of neoplastic cells [131].

### 2.13. Malignant Cells

The production of autocrine growth factors that activate pro-oncogenic pathways is observed in Sézary cells. Malignant cells are influenced by IL-4 and IL-13, which activate the STAT6 pathway. This pathway will enhance the transcription of IL-4, IL-5, and IL-13 messenger RNA, and will lead to the secretion of these cytokines by the malignant cells [15,123].

The IL-15 activates STAT3 and STAT5, and it acts as a tumor growth factor [132]. In neoplastic cells, ZEB1, a tumor suppressor gene that represses the transcription of IL-15, is hypermethylated. This impairs ZEB1 function, IL-15 is overexpressed and acts in an autocrine manner stimulating tumor cell growth. The IL-15 secreted by malignant cells also influences the microenvironment, especially the epidermal keratinocytes, that become activated and proliferate [15,133].

The IL-32 also acts in an autocrine manner. It is secreted by malignant cells and stimulates tumor growth by the NF-κB pathway activation [15,134].

Cyclooxygenases (COX) are enzymes that mediate inflammation through the conversion of arachidonic acid to prostaglandin. The COX-2 is detected in malignant cells. It increases the production of prostaglandin E2 (PGE2), which in turn reduces cell-mediated immunity by inhibiting Th1 cytokine production and suppressing NK and CD8 T-cell cytotoxicity. The PGE2 may also bind to its receptor on the Sézary cell surface and promote malignant cell growth [15,135].

Besides the modulation of the inflammatory microenvironment, malignant cells may suppress and kill reactive cells by cell-to-cell contact. The JAK-STAT pathway may induce the expression of CD80 and PD-L1 on malignant cells. The CD80 binds to the CTLA-4 and PD-L1 binds to the PD-1 on inflammatory lymphocytes, inhibiting their function and favoring immune evasion [15,21,136].

Malignant cells express FasL that induces apoptosis upon binding to Fas on inflammatory CD8+ T-cells. On the other hand, malignant cells show fewer Fas expressions and are resistant to Fas-L-mediated apoptosis [15,137].

## 3. Angiogenesis and Lymphangiogenesis

The formation of vascular and lymphatic vessels contributes to the dissemination of malignant T-cells [30]. Neo-angiogenesis and lymphangiogenesis are analyzed by micro-vessel density (expression of matrix metalloproteinases 2 and 9, and CD34) or by VEGF expression (VEGF-A for angiogenesis and VEGF-C for lymphangiogenesis) [14]. In CTCL skin lesions, there is an increased number of microvessels. Moreover, VEGF is significantly expressed on these cells, and it is produced by neoplastic cells, endothelial cells, histiocytes, fibroblasts, and reactive T-cells [138,139,140].

Malignant T-cells produce VEGF-A by JAK and c-Jun N-terminal kinase (JNK)-dependent mechanisms. The VEGF-A is a potent angiogenesis stimulator, and it also induces the expression of TSLP in keratinocytes [30,141]. Malignant T-cells also produce other pro-angiogenic factors such as IL-17F, angiopoietin-2, placental growth factor, and YKL-40 [30]. The increase in podoplanin+ lymphatic vessels is caused by the release of VECG-C. Its density is associated with disease progression [30,142]. Lymphotoxin-α (LTα) is involved in lymphatic and secondary lymphoid structures formation. It is expressed in CTCL by a JAK3/STAT5 pathway. LTα acts in an autocrine manner by stimulating the expression of IL-6 in malignant cells. The LTα, IL-6, and VEGF promote angiogenesis, which will ultimately contribute to tumor growth and spread [14].

Matrix metalloproteinases (MMPs) are a group of enzymes involved in diverse physiologic (tissue remodeling, embryogenesis) and pathologic (autoimmune diseases, cancer) functions [143]. They have proteolytic activity, cellular growth signaling, apoptosis regulation, angiogenesis, and they participate in inflammatory pathways [144]. The MMPs function is regulated by its inhibitors, tissue inhibitors of metalloproteinases (TIMPs) [143,145]. The MMPs are produced by fibroblasts, macrophages, keratinocytes, T-cells, endothelial cells, and mast cells [146]. In CTCL skin, MMP-2 and MMP-9 are increased. Malignant T-cells secrete factors that stimulate the production of MMPs by stroma cells [30]. These two metalloproteinases are upregulated in different types of cancer, and they mediate microvascular proliferation by degrading basement membranes of endothelial cells, facilitating the spread of proliferating vascular cells to the surrounding stroma [138,146,147]. On the other hand, in a rabbit study model, the active form of MMP-5 was reduced in malignant T-cells [148]. Thus, the role of different MMPs in CTCL is still not fully understood.

## 4. Skin Barrier

### 4.1. Staphylococcus aureus

Sézary syndrome skin lesions and nasal cavity display a higher incidence of *Staphylococcus aureus* (*S. aureus*) colonization [149]. This finding is similar to atopic dermatitis (AD). In SS, there is impaired production of antimicrobial peptides, including cathelicidins and β-defensins [150]. The staphylococcal enterotoxins (SEs) function as superantigens and activate STAT3 and induce IL-10 and IL-17 expression on neoplastic cells. The IL-10 is an immunosuppressive cytokine; however, if IL-17 contributes to antimicrobial defense and/or lymphomagenesis is a matter of debate [26]. The SE also triggers the expression of FoxP3 by Sézary cells in a STAT5-dependent manner, but it is not defined whether the expression of this Treg marker has any potential immune-regulatory effect produced by malignant cells [49]. The miR-155 is also STAT5-dependent, and it inhibits the Th1 STAT4 transcription factor, and SEs may contribute to the expression of this microRNA in SS (Figure 4 and Figure 10) [27]. The malignant T-cells initially induce susceptibility towards *S. aureus* and, subsequently, initiate crosstalk between benign and malignant T-cells, resulting in the activation of pro-oncogenic pathways [15].

### 4.2. Galectins

Galectins are a family of soluble carbohydrate-binding proteins with intra and extracellular activity defined by the carbohydrate specificity and the galectin structure [151]. These lectins have been described as players in a wide variety of cellular processes crucial in immune functions and cancer progression, such as the induction of angiogenesis, resistance to apoptosis, continuous cell proliferation, cytokine secretion, and chemotaxis [152]. Not surprisingly, there are several reports of galectin involvement in many cancers, including hematological malignancies [152].

Tumor-derived galectin-1 (Gal-1) inhibits proliferation and Th1 cytokine production by nonmalignant T-cells, besides inducing Th2 cytokines and the suppression of antitumor immune responses [153]. Gal-1 also induces epidermal hyperproliferation and impairs epidermal barrier function due to the loosely packed desmosomes, which explains the increased incidence of skin infections [13,154]. Gal-1 is known also by inducing apoptosis on T-cells [155], but the lack of CD7 expression and different CD7 glycosylation in tumor cells on skin lesions and Sézary T-cell lines seems to confer Gal-1 induced apoptosis resistance to the tumor cells [156,157].

Although galectin-9 (Gal-9) is highly expressed on lesional skin, in the serum and secreted in higher amounts by patients’ tumor cells [155], high doses of exogenous Galectin-9 induce apoptosis on CTCL cell lines in vitro and reduce T-cell tumor formation on a murine model [158].

## 5. Conclusions

Most studies about the physiopathology of MF/SS focus on the evaluation of malignant cells. Different genetic and epigenetic alterations in SS are described and, despite the heterogeneous findings, they converge to the JAK/STAT pathway alterations, with an increased STAT3, STAT5, and STAT6 and decreased STAT4 activation. The shift from a Th1 to a Th2 immunologic response is driven by complex interactions between malignant and tumor microenvironment cells. These interactions are mediated by cytokines, chemokines, growth factors, transcription factors, and other molecules produced by a myriad of different cells. Thus, strong evidence points towards the important role of the inflammatory pro-tumorigenic environment that contributes to the immune escape of Sézary cells. Understanding the mechanisms that contribute to tumor growth and inhibition of anti-tumor response will contribute to the search for more effective treatments that not only kill malignant cells but also reestablish the normal immunologic milieu. Maybe, the combination of tumor-directed and immune-enhancing therapies may present a perspective of long-term control and even a cure for this complex and life-threatening disease.

## Figures and Tables

**Figure 1 ijms-23-00936-f001:**
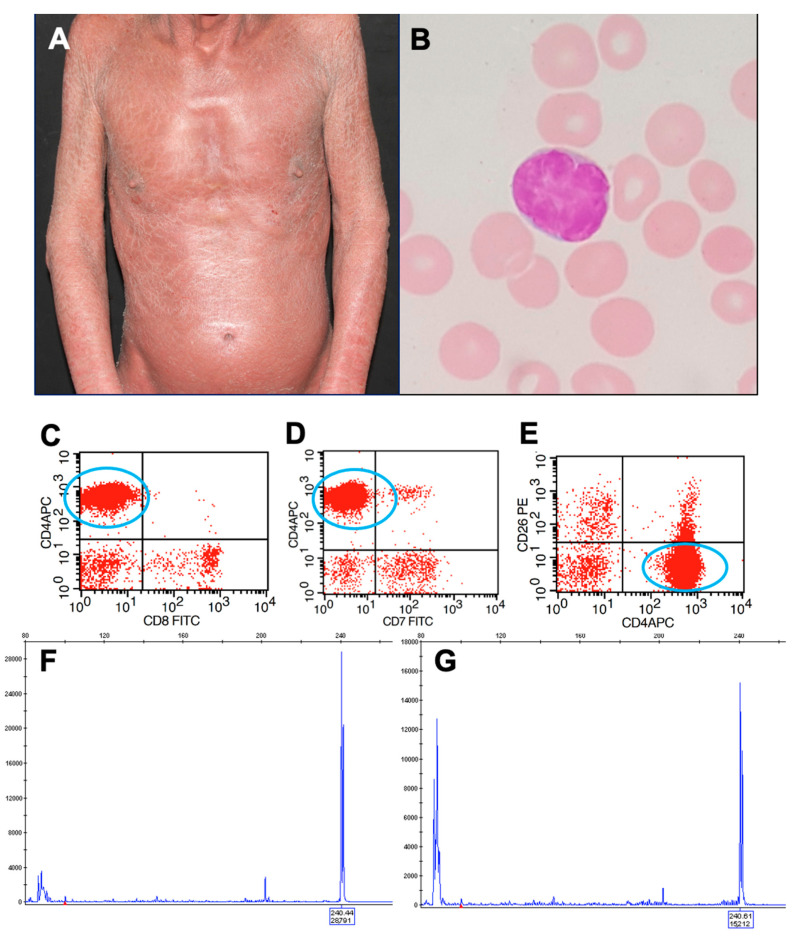
Sézary syndrome. Erythroderma (**A**). Sézary cell on a peripheral blood smear (**B**). Flow cytometry of peripheral blood showing the CD4+CD8− (**C**), CD4+CD7− (**D**), and CD4+CD26− (**E**) Sézary cells. Monoclonal T-cell population on the skin (**F**) and the same clone detected on peripheral blood (**G**).

**Figure 2 ijms-23-00936-f002:**
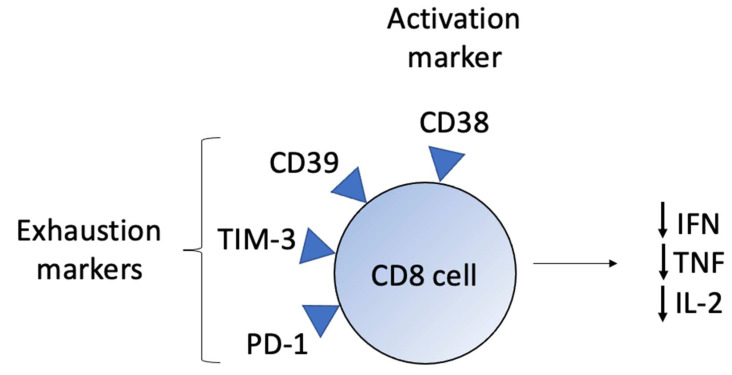
CD8+ T-cell characteristics in SS. Activation of CD8+ T-cells is detected by the expression of CD38, and the chronic activation leads to an exhaustion phenotype with the expression of CD39, TIM-3, and PD-1. Furthermore, in exhausted CD8+ T-cells, a functional loss is observed, with reduced production of IFN, TNF, and IL-2, decreasing the cytotoxic activity against neoplastic cells.

**Figure 3 ijms-23-00936-f003:**
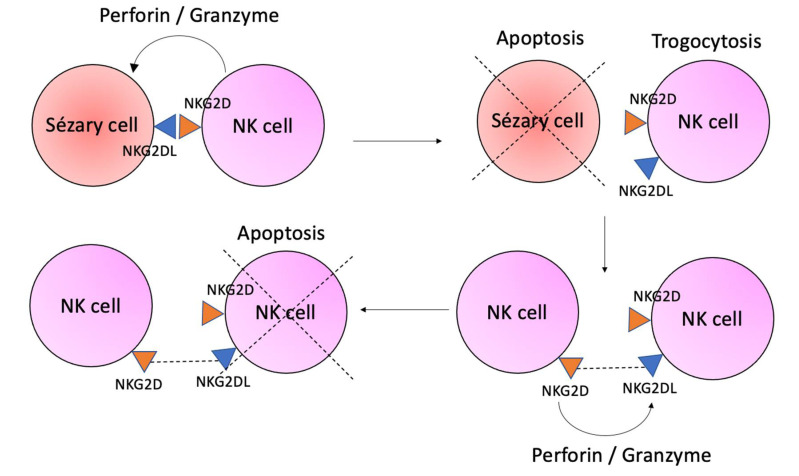
Trogocytosis process. The interaction of Sézary cells and NK cells by NKG2D/NKG2DL activates the degranulation of perforin and granzyme that kills malignant cells. The cell-to-cell contact allows the migration of NKG2DL from the Sézary cell to the NK cell in the trogocytosis process. The NK cell with NKG2DL is recognized by tumor-naïve NK cells that liberate perforin and granzyme, killing the NK cell.

**Figure 4 ijms-23-00936-f004:**
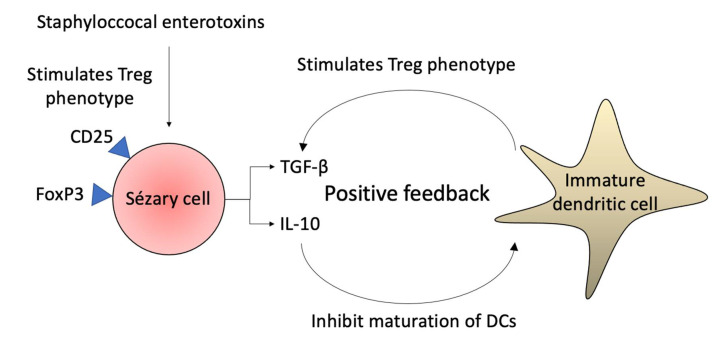
Sézary cell and dendritic cell interaction. The Sézary cells express a Treg phenotype upon Staphylococcal enterotoxins stimulation. The cytokines produced by malignant cells, especially IL-10, prevent DCs from maturating; and the immature DCs promote immune tolerance and persistence of Treg phenotype in Sézary cells in positive feedback.

**Figure 5 ijms-23-00936-f005:**
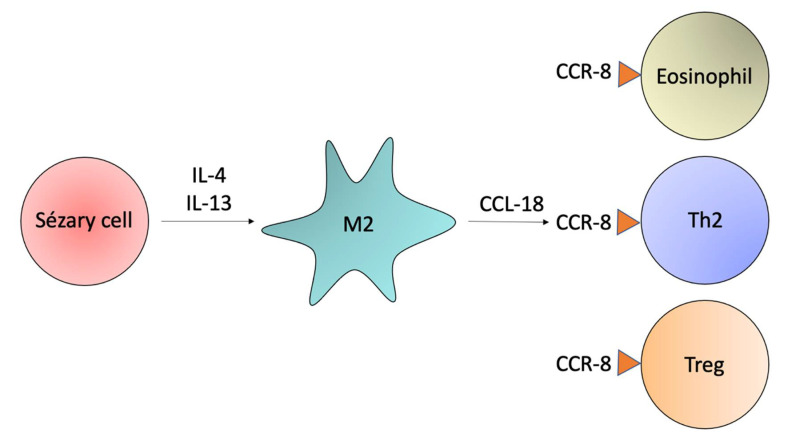
M2 macrophages. Sézary cells produce the Th2 cytokines IL-4 and IL-13. These cytokines stimulate M2 macrophages to produce CCL18, a chemokine that attracts CCR8+ cells to the tumor microenvironment. The CCR8 is expressed by eosinophils, Th2 cells, and Treg.

**Figure 6 ijms-23-00936-f006:**
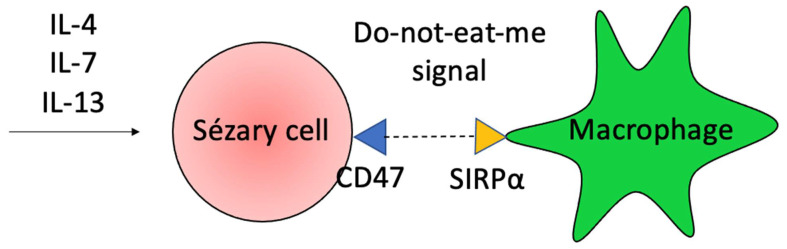
Do-not-eat-me signal. The Th2 microenvironment stimulates the expression of CD47 on the Sézary cell surface. It binds to SIRPα on macrophages and prevents phagocytosis, contributing to tumor escape.

**Figure 7 ijms-23-00936-f007:**
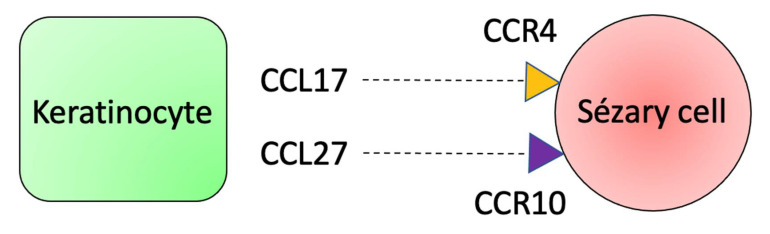
Keratinocytes produce CCL17 and CCL27 that bind to CCR4 and CCR10, respectively. These receptors are expressed by Sézary cells, contributing to the epidermotropism of malignant cells.

**Figure 8 ijms-23-00936-f008:**
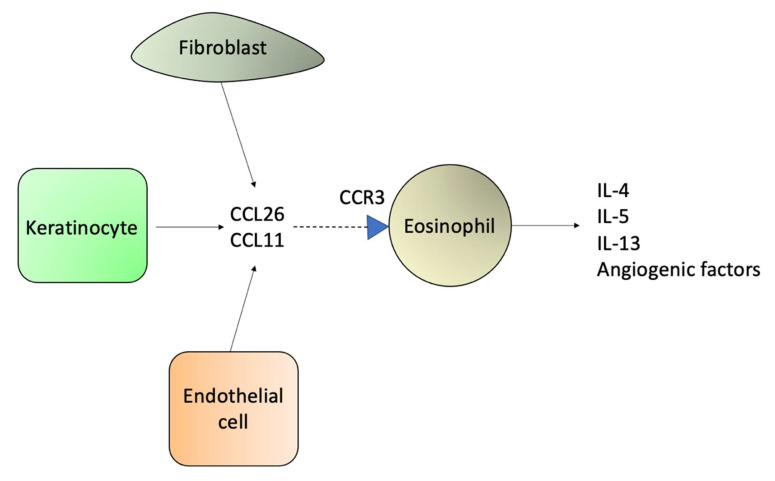
Keratinocytes, fibroblasts, and endothelial cells produce CCL26 and CCL11. These chemokines attract CCR3+ eosinophils.

**Figure 9 ijms-23-00936-f009:**
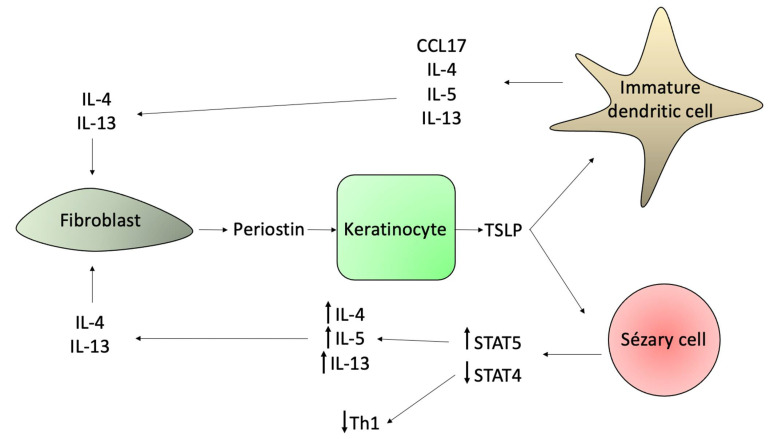
The role of periostin on the SS tumor microenvironment. The periostin, secreted by fibroblasts after activation by IL-4 and IL-13, stimulates TSLP production by keratinocytes. The TSLP stimulates immature DCs and Sézary cells to secrete Th2 cytokines, which will ultimately interact with fibroblasts to produce more periostin in a positive loop.

**Figure 10 ijms-23-00936-f010:**
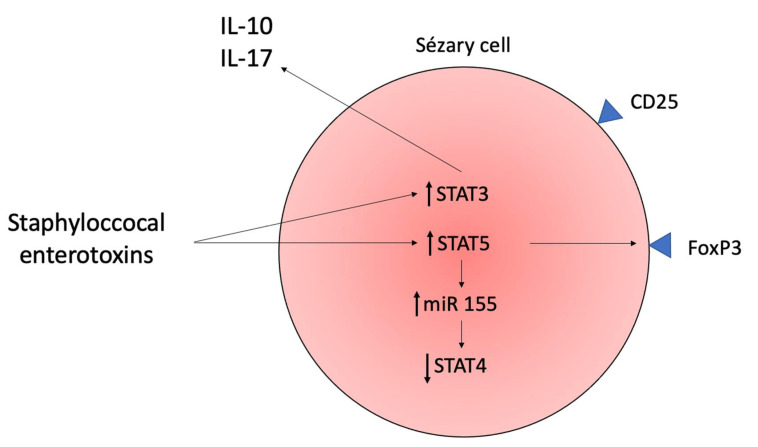
The interaction of *Staphylococcus aureus* enterotoxins in Sézary cells. The SE superantigen increases STAT3 and STAT5 function. The STAT3 pathway leads to increased production of IL-10 and IL-17. The IL-10 prevents DCs from maturating and favors the expression of the Treg phenotype in Sézary cells. The STAT5 increases production of miR-155 that inhibits the Th1 STAT4 pathway.

## Data Availability

Not applicable.

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
