# Peer review of "The Role of Tumor Microenvironment in the Pathogenesis of Sézary Syndrome"

_ijms, 2022, doi:10.3390/ijms23020936_

Round 1
Reviewer 1 Report
This review is aimed to summarize the current knowledge of the role of tumor microenvironment in the pathogenesis of Sézary syndrome. The paper was not well written. The role of tumor microenvironment in the pathogenesis of Sézary syndrome should be comprehensively reviewed, and the content should be well organized and presented. The lack of deep analysis and summary is a major weakness. The overall quality of this review paper is not very high. The major revision is needed.
Author Response
Dear reviewer, thank you for your considerations. We discussed the topics more extensively to provide a comprehensive review of the current literature. English was edited and corrected by a professional reviewer.
Reviewer 2 Report
Miyashiro and colleagues presented an interesting review article on Sézary syndrome focusing their attention mainly on the role of the surrounding cells and tumors microenvironment. Overall, the review is well structured, however, the description of some parts should be improved. Please see the comment below:
The present review is a description of the role of different cell populations involved in SS rather than a description of the role of the tumor microenvironment. The authors have to better elucidate the molecular mechanisms responsible for tumor cell invasion and diffusion as well as the mechanisms of SS recurrence. For example, the authors only mention the role of MMPs in the above-mentioned mechanisms as well as the mechanisms behind angiogenesis and lymphatic distribution of malignant cells. Different studies have described the role of MMPs in these processes in different cutaneous tumors and T-cell lymphomas. Please add these data. For this purpose, see:
- PMID: 32392801
- PMID: 17089123
- PMID: 33352267
Author Response
Dear reviewer, thank you for your considerations. We discussed the topics more extensively to provide a comprehensive review of the current literature. The suggested articles were added to the references. English was edited and corrected by a professional reviewer.
Round 2
Reviewer 1 Report
This revised review is not significantly improved. The paper is not well written. The role of tumor microenvironment in the pathogenesis of Sézary syndrome should be comprehensively reviewed, and the content should be well organized and presented. However, this review paper is largely an accumulated relevant literature. The lack of deep analysis and summary is a major weakness. The overall quality of this review paper is not very high.
The followings are a few examples of pitfalls.
Where are the original data published for Figure 1? If it is not published data, who generated these data? A general description of the data should be provided. Although the authors provided the reference 2, 6, and 7, the data showed in Figure 1 are not presented in these three referenced papers.
The font size in Figure 1F and 1G are so small that it is difficult to read it.